# Preliminary Evaluation of NT-proBNP and cTnI as Predictors of Procedure Safety in Dogs Undergoing Transcatheter Edge-to-Edge Mitral Valve Repair

**DOI:** 10.3390/vetsci12030223

**Published:** 2025-03-02

**Authors:** Jeong-Min Lee, Seung-Keun Lee, Kyoung-A Youp, Ah-Ra Lee, Young-Wook Cho, Youn-Seo Jung, Sun-Tae Lee

**Affiliations:** Korea Animal Medical Center, Cheongju 28651, Republic of Korea; memo76777@gmail.com (J.-M.L.); koreavet@hanmail.net (S.-K.L.); y01032002258@gmail.com (K.-A.Y.); alvetrad09@gmail.com (A.-R.L.); duddnr0509@gmail.com (Y.-W.C.); dbsseo5046@gmail.com (Y.-S.J.)

**Keywords:** canine, myxomatous mitral valve disease, mitral valve repair, transcatheter edge-to-edge repair

## Abstract

Transcatheter edge-to-edge mitral valve repair (TEER) is an emerging minimally invasive treatment for dogs with myxomatous mitral valve disease (MMVD). This study evaluated pre-procedure N-terminal pro-brain natriuretic peptide (NT-proBNP) and cardiac troponin I (cTnI) levels to assess their association with procedure safety. Additionally, the relationship between pre-procedure NT-proBNP and cTnI levels and the length of hospitalization was analyzed in the survival group. While the procedure safety supports the application of this technique as a potential therapeutic option in the short term, no comparative observations can be made at this time from the results of this investigation.

## 1. Introduction

Myxomatous mitral valve disease (MMVD) is the most common cause of congestive heart failure and the most prevalent acquired cardiac disease in dogs [1]. Auscultation, electrocardiography, thoracic radiography, and echocardiography are commonly used to diagnose MMVD [2]. In addition, cardiac biomarkers such as N-terminal pro-B-type natriuretic peptide (NT-proBNP) and cardiac troponin I (cTnI) serve as to assess the functional response to the MMVD and help define the disease severity [1,3,4]. Among these, NT-proBNP is a peptide produced by the heart, particularly the ventricles, in response to increased cardiac pressure or strain [5]. Additionally, cTnI serves as a biomarker that reflects the severity of myocardial injury [6].

Transcatheter edge-to-edge mitral valve repair (TEER) is an emerging treatment option for canine MMVD [7,8,9]. It is a minimally invasive interventional mitral valve repair technique that has been increasingly adopted and is now used in the US, Europe, and Asia [8,9,10]. In 2020, a preliminary study on TEER in eight dogs with MMVD stage B1 demonstrated its effectiveness in significantly reducing mitral regurgitation [8]. More recently, a study involving 50 dogs with MMVD stages B2, C, and D reported a procedure feasibility rate of 96% and a significant decrease in regurgitation volume [9]. Assessing which patients are at higher risk before the procedure is crucial for clinical decision making and discussions with owners. In humans, higher preoperative NT-proBNP levels have been identified as a risk factor for perioperative mortality at follow-up [11]. Elevated pre-procedure baseline cardiac troponin I levels have also been shown to be associated with significantly higher 1-year mortality compared to lower values [12]. However, as TEER has only recently been introduced in veterinary medicine, no studies have yet evaluated its association with survival to hospital discharge as a measure of procedure safety in dogs. Therefore, this preliminary study assessed the association between pre-procedure NT-proBNP and cTnI levels with procedure safety following TEER in dogs with MMVD.

## 2. Materials and Methods

### 2.1. Animals

The medical records of all dogs treated with TEER at the Korea Animal Medical Center between September 2023 and January 2025 were electronically reviewed for suitability. The study included dogs with MMVD stage C or D, classified according to the American College of Veterinary Internal Medicine MMVD consensus guidelines. All of the dogs were receiving pimobendan and furosemide before undergoing TEER. Data on comorbid conditions, such as chronic kidney disease (CKD) staged according to the International Renal Interest Society guideline, were also collected to provide a comprehensive understanding of the study population [13]. Only dogs with either cTnI or NT-proBNP measurements taken within 7 days before the TEER procedure were included in the study. Regardless of comorbidities, the dogs underwent TEER if the attending clinician determined that the procedure was recommended for managing their MMVD. Dogs with incomplete measurements or measurements taken outside the specified time points were excluded from the study.

### 2.2. Measurement of the Blood Concentration of NT-proBNP and cTnI

Dogs underwent a series of screening tests to evaluate TEER’s potential benefits and risks. The tests included the complete blood count, serum chemistry, venous blood gas analysis, electrocardiography, thoracic radiographs, transthoracic echocardiography, transesophageal echocardiography (TEE), NT-proBNP, and cTnI. All of the dogs were fasted for 10 h before blood collection. Blood samples were obtained from the jugular or cephalic veins during the pre-TEER screening. The collected serum samples were analyzed within 1 h of collection using the Vcheck Canine NT-proBNP Test kit (Bionote, Hwaseong, Republic of Korea) and Vcheck Canine TnI Test kit (Bionote, Hwaseong, Republic of Korea), following the manufacturer’s instructions.

### 2.3. TEER Procedure

The TEER device used in this study was the ValveClamp (Hongyu Medical Technology, Shanghai, China), with the procedural steps guided by techniques previously described [8,9]. In brief, all of the dogs were induced with propofol and maintained under general anesthesia throughout the procedure. An intra-arterial catheter was placed to monitor arterial pressure during the intervention. TEE, incorporating three-dimensional real-time multiplanar reconstruction and simultaneous orthogonal biplane views, was employed to assess and visualize the mitral valve and determine its precise location. A minimal incision was made in the pericardium for the transapical puncture required for the TEER. Using TEE guidance, left atrial steering and clip positioning were evaluated to ensure proper alignment and trajectory of the TEER clamp device. Once optimal access was established, the clip delivery system was introduced, and alignment was confirmed through 3D TEE. Leaflet grasping was carried out under TEE guidance, followed by the deployment of the TEER clamp device. After the procedure, all of the dogs were transferred to the intensive care unit for close post-procedure monitoring.

### 2.4. Postoperative Follow up

The survival group was defined as dogs that were discharged after the TEER procedure, while the non-survival group consisted of dogs that either died or were euthanized before discharge. Procedure safety was defined as survival to hospital discharge based on previous study. Generally, hospital discharge was determined at the clinician’s discretion, based on the dogs’ progressive improvement, alertness, normal appetite, stable vital signs, and absence of cardiovascular deterioration. In the survival group, the time from the completion of the procedure to discharge was defined as the length of hospitalization. To evaluate the procedure efficacy, echocardiographic assessments, including the left atrium-to-aortic root ratio (LA/AO) and left ventricular end-diastolic diameter normalized for body weight (LVIDdN), were performed by a single operator on post-procedure day 3.

### 2.5. Statistical Analysis

Statistical analyses were performed using SPSS software (version 29.0; IBM Corp, Chicago, IL, USA). The Shapiro–Wilk test was employed to assess whether the research variables followed a normal distribution. Variables with a normal distribution were described using the mean and standard deviation, while those with a non-normal distribution were reported as the median and range. Homogeneity between groups was evaluated using independent *t*-tests and Fisher’s exact tests. Furthermore, Pearson’s correlation coefficient was used to examine the relationship between the length of hospitalization, cTnI, and NT-proBNP levels. A *p*-value of less than 0.05 was considered statistically significant.

## 3. Results

### 3.1. Characteristics of the Study Population

In total, 25 dogs were included in the study. The median age was 11 years (range: 9–14 years). The study population consisted of 12 spayed females, 12 castrated males, and 1 intact male. The median body weight was 4.3 kg (range: 2.1–7.8 kg). The most common breeds among the dogs in the study were Maltese (n = 7; 28%), mixed breed (n = 5; 20%), Schnauzer (n = 3; 12%), Shih Tzu (n = 3; 12%), Poodle (n = 3; 12%), Pomeranian (n = 2; 8%), followed by Chihuahua and Dachshund (n = 1 each; 4%). Among the 25 dogs, 19 were classified into the survival group and 6 into the non-survival group. Of the survival group, 15 were diagnosed with MMVD stage C and 4 with MMVD stage D. In the non-survival group, three were diagnosed with MMVD stage C and three with MMVD stage D. Detailed baseline characteristics are presented in Table 1.

### 3.2. Concurrent Diseases

Among the 25 dogs included in the study, 23 (92%) had comorbid conditions. In the survival group, 18 out of 19 dogs (94.7%) had concurrent diseases, while, in the non-survival group, 5 out of 6 dogs (83.3%) had comorbidities. In the survival group (n = 19), comorbidities included hyperadrenocorticism (n = 6), tracheal collapse (n = 5), urolithiasis (n = 3), gallbladder mucocele (n = 1), chronic pancreatitis (n = 1), hydrocephalus (n = 1), intestinal leiomyosarcoma (n = 1), esophageal stricture (n = 1), brain tumor (n = 1), liver tumor (n = 1), peritoneopericardial diaphragmatic hernia (n = 1), idiopathic epilepsy (n = 1), and meningoencephalitis of unknown etiology (n = 1). In the non-survival group, two dogs had CKD stage 2 and two had CKD stage 1. Other comorbidities in this group included bronchial collapse (n = 1), urolithiasis (n = 1), and bacterial cystitis (n = 1).

### 3.3. Association of NT-proBNP and cTnI Levels with Survival

NT-proBNP was measured in 25 dogs, while cTnI was measured in 19 dogs, including 15 in the survival group and 4 in the non-survival group. In cases where NT-proBNP was measured but cTnI was not, the decision was primarily influenced by cost constraints. The median NT-proBNP level for all patients was 1870.1 pg/mL (range: 500–10,000). In the survival group, the median NT-proBNP was 1262.6 pg/mL (range: 500–8773), while, in the non-survival group, it was 3557 pg/mL (range: 774–10,000). No significant difference in preoperative NT-proBNP levels was observed between the survival and non-survival groups (*p* = 0.187), as presented in Figure 1. The median cTnI level for all patients was 0.21 ng/mL (range: 0.01–3.6). In the survival group, the median cTnI was 0.1 ng/mL (range: 0.01–3.6), while, in the non-survival group, it was 0.39 ng/mL (range: 0.22–0.51). Similarly, no significant difference in preoperative cTnI levels was found between the two groups (*p* = 0.869), as presented in Figure 2. In the survival group, the median length of hospitalization was 4 days (range: 3–30 days). Both NT-proBNP and cTnI levels showed no significant correlation with the length of hospitalization, or with NT-proBNP (r = −0.255; *p* = 0.291) and cTnI (r = −0.151; *p* = 0.623). Among the patients with prolonged hospitalization, two were hospitalized for 24 and 30 days, respectively. Both developed acute kidney injury after TEER and required extended hospitalization for management. A significant reduction in the left atrium-to-aortic root ratio (2.20 [1.72–2.78] vs. 1.93 [1.27–2.75], *p* < 0.026) and left ventricular end-diastolic diameter normalized for body weight (2.11 [1.77–2.45] vs. 1.87 [1.56–2.17], *p* < 0.001) was observed in the surviving dogs following TEER.

## 4. Discussion

To the best of our knowledge, this retrospective study is the first to evaluate the association between preoperative levels of biomarkers (NT-proBNP and cTnI) and clinical outcomes following the TEER of MMVD dogs. In humans, elevated baseline troponin levels have been shown to correlate with in-hospital mortality as well as short-term and long-term mortality in acute heart failure [14,15]. Furthermore, higher cardiac troponin levels are well-documented predictors of worse clinical outcomes [12]. However, in this study, NT-proBNP and cTnI were not found to have a significant association with procedure safety, suggesting that these biomarkers may not be of value in the selection of suitable surgical candidates.

In this study, 6 out of 25 dogs died, resulting in an observed survival rate to hospital discharge of 76%. In contrast, a previous study reported a TEER procedure safety rate of 96%, with the survival to hospital discharge observed in 48 out of 50 dogs with MMVD across stages B2, C, and D [9]. This discrepancy may reflect differences in the inclusion and exclusion criteria, as the previous study excluded cases with severe pulmonary hypertension (tricuspid regurgitation pressure gradient > 80 mmHg), non-cardiac diseases likely to affect six-month survival, and evidence of active systemic infection or immune-mediated disease, which were not considered in our study [9]. Additionally, the severity of disease in the study populations may have differed; in our study, 28% of patients were classified as MMVD stage D, whereas only 6% were classified as such in the previous study [9]. TEER using Mitraclip^®^ is primarily employed in human patients with mitral valve regurgitation who are considered high-risk candidates for open-heart surgery, and evidence suggests that increased procedural experience is associated with improved clinical outcomes [16]. The importance of the operator’s experience has also been emphasized in cases of canine MMVD [9]. Additionally, differences in comorbid conditions between studies may have played a role. In our study, a high proportion of dogs presented with concurrent diseases such as CKD, bronchial collapse, or other systemic illnesses, which may have impacted their prognosis and survival outcomes. The small size of the dogs and the relatively high proportion of MMVD stage D cases in our cohort could have further contributed to the lower survival rate. These findings highlight the need for future studies to consider the impact of comorbidities on TEER outcomes in dogs with MMVD.

Our study observed a higher prevalence of comorbid conditions in the survival group compared to the non-survival group. Although comorbidities were more frequent in the survival group, their influence on procedure safety remains unclear, which may be partially attributed to the small sample size. These findings suggest that factors beyond MMVD, such as comorbid conditions, may have influenced the preoperative cTnI and NT-proBNP levels in these dogs. In humans, elevated cTnI levels are associated with various non-cardiac conditions, including severe renal failure, diabetes, and critical illness, though the exact mechanisms remain unclear [17,18]. Similarly, increased cTnI concentrations are commonly observed in dogs and cats with azotemic renal failure or other systemic non-cardiac illnesses [18]. This elevation may result from myocardial cell injury caused by factors such as cardiac ischemia, toxicity, remodeling, trauma, or inflammation, leading to the release of cTnI from the myocardium [19]. Additionally, plasma NT-proBNP concentrations are known to increase in dogs with azotemia [20]. Therefore, caution is warranted when interpreting the results of this study, as these comorbid conditions may have influenced the biomarker levels observed.

In the present study, both NT-proBNP and cTnI showed no significant association with the length of hospitalization. This may be attributed to the small sample size of our study or variations in the hospitalization duration influenced by the patients’ underlying conditions. Notably, NT-proBNP is recognized as an important marker of hemodynamic and functional characteristics and has been shown in human studies to predict longer intensive care unit stays, including a demonstrated association between cardiac biomarkers and the hospital length of stay in pediatric patients [21]. In this study, the median length of hospitalization was 4 days, which was longer than the median of 2 days reported in previous studies involving MMVD patients undergoing TEER [9]. Among the survival group, two dogs developed acute kidney injury (AKI), with hospitalization durations of 24 and 30 days, respectively. These outliers may have contributed to the observed difference in the hospitalization length. In humans, AKI occurred in 14.7% following MitraClip procedures [22]. However, this study reported no significant difference in the hospitalization length between patients with and without AKI [22]. Further research is needed to investigate complications following TEER and their impact on the hospitalization duration.

This study was an observational, retrospective, single-center study, and the relatively small sample size limits the ability to draw firm conclusions. Additionally, our study included patients with multiple comorbidities, which were not excluded. These comorbid conditions may have influenced the outcomes, potentially affecting the interpretation of our findings. Additionally, the results may be influenced by an initial learning curve, which could have impacted the consistency of the measurements or interpretations. Further studies with larger sample sizes and more standardized methodologies are necessary to validate these findings.

## 5. Conclusions

No significant association was observed between these biomarkers and short-term survival; the results underscore the complexity of predicting outcomes in this population. Future studies should focus on larger cohorts with multi-centers and long-term follow-up to better understand the interplay of biomarkers, procedural factors, and patient-specific characteristics. Additionally, the influence of comorbidities on preoperative biomarker levels and survival outcomes should be explored in greater detail to enhance the clinical utility of these measurements in guiding individualized decision making for TEER in canine MMVD.

## Figures and Tables

**Figure 1 vetsci-12-00223-f001:**
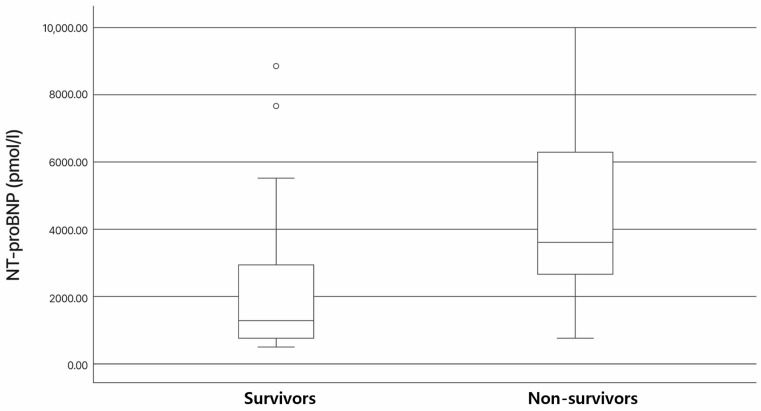
Box and whisker plots comparing N-terminal pro-brain NP (NT-proBNP) in dogs with myxomatous mitral valve disease stages C and D receiving transcatheter edge-to-edge mitral valve repair. The box represents the interquartile range (IQR, 25th–75th percentile), the horizontal line within the box indicates the median, and the whiskers extend to the minimum and maximum values, excluding outliers. Median NT-proBNP was lower in the survival group than in the non-survival group. However, no significant differences were found in NT-proBNP between the two groups.

**Figure 2 vetsci-12-00223-f002:**
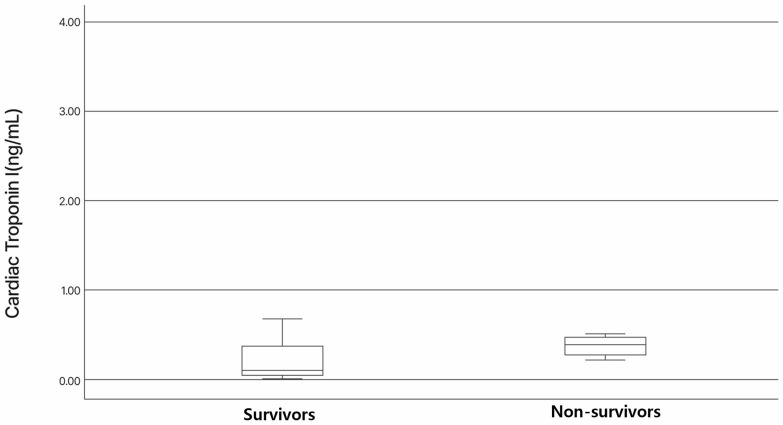
Box and whisker plots comparing cardiac troponin I (cTnI) in dogs with myxomatous mitral valve disease stages C and D receiving transcatheter edge-to-edge mitral valve repair. The box represents the interquartile range (IQR, 25th–75th percentile), the horizontal line within the box indicates the median, and the whiskers extend to the minimum and maximum values, excluding outliers. Median cTnI was lower in the survival group than in the non-survival group. However, no significant differences were found in cTnI between two groups.

**Table 1 vetsci-12-00223-t001:** Demographics and baseline characteristics of dogs undergoing transcatheter edge-to-edge mitral valve repair procedures.

Variable	Total	Survivals	Non-Survivals
Gender			
Male	1 (4%)	1 (5.2%)	0 (0%)
Neutered male	12 (48%)	8 (42.1%)	4 (66.6%)
Female	0 (0%)	0 (0%)	0 (0%)
Neutered female	12 (48%)	10 (52.6%)	2 (33.3%)
Age (year)	11 (9–14)	10 (9–14)	12 (9–13)
Weight (kg)	4.3 (2.1–7.8)	4.6 (2.3–7.8)	2.9 (2.1–6.0)
Breed			
Maltese	7 (28%)	4 (21.1%)	3 (50%)
Mix-breeds	5 (20%)	4 (21.1%)	1 (16.7%)
Schanuzer	3 (12%)	3 (15.8%)	0 (0%)
Shih Tzu	3 (12%)	3 (15.8%)	0 (0%)
Poodle	3 (12%)	3 (15.8%)	0 (0%)
Pomeranian	2 (8%)	1 (5.3%)	1 (16.7%)
Chihuahua	1 (4%)	1 (5.3%)	0 (0%)
Dachshund	1 (4%)	0 (0%)	1 (16.7%)
Total	25	19	6
N (%), Median (Range)			

## Data Availability

The original contributions presented in the study are included in the article; further inquiries can be directed to the corresponding author.

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
