# Peer review of "Preliminary Evaluation of NT-proBNP and cTnI as Predictors of Procedure Safety in Dogs Undergoing Transcatheter Edge-to-Edge Mitral Valve Repair"

_vetsci, 2025, doi:10.3390/vetsci12030223_

Round 1

Reviewer 1 Report

Comments and Suggestions for Authors

vetsci-3421891

This paper retrospectively examines N‐terminal pro‐brain natriuretic peptide (NT‐proBNP) and cardiac troponin I (cTnI) levels collected preoperatively in a convenience sample of dogs undergoing TEER for management of myxomatous mitral valve disease (MMVD). The stated objectives were to retrospectively evaluate the possible value of the biomarkers as prognostic indicators prior to the procedure by looking for any association with 30-day survival and also to examine the association, if any, with the length of hospital stay. I am not sure these objectives have been adequately achieved.

I am concerned that the authors may be attempting to make too much of too little. Much of their data are contradictory, insufficient information is provided to fully assess and interpret their findings or to make adequate comparisons with previous studies, and outcomes and endpoints are inadequately defined. It may be wise to gather more information both from previous cases and from future cases in order to perform a more in-depth analysis better able to guide the application of the technique, along the lines of the TEER studies they cite. These perspectives are implicit in most of the comments that follow.

Simple summary. Line 16. Neither of these outcomes were achieved by this study, which neither evaluated the technique of TEER in relation to other treatments nor generated information for immediate application in guiding procedural decisions. A more accurate statement would be that "while the 30 day survival rate supports the application of this technique as a potential therapeutic option in the short term, no comparative observations can be made at this time from the results of this investigation. Results of the present study did not reveal potential for use of the selected biomarkers in pre-procedure assessment”. Comments made in this summary also do not reflect those appearing in the abstract.

Line 22. Change "between" to "of".

It is appropriate to confirm that animals subjected to this procedure survive the immediate procedure and a 30 day horizon is reasonable. However, surely it is equally important to confirm that there is a clinical improvement in treated patients and also to compare long-term survival and to demonstrate at least equivalent survival and quality of life for treated and untreated animals. I am a little surprised that these issues do not receive any attention, especially since the tested biomarkers appear to have little prognostic value. Do not all of these concerns have equivalent importance?

Line 45. This paragraph is a little confused. The biomarkers studied in this investigation are not used to diagnose MD but to assess functional response to the disease and evidence of failure and help define disease severity.

Line 52."… that has been increasingly…".

Line 57. "Risk" - please define what is meant by risk. Presumably you could be referring to risk for an adverse response to the procedure or risk for death if the procedure is withheld, among others. This ambiguity concerning what the authors see as the primary value of performing biomarker assays is recurrent throughout the paper. This extends to the question of "prognosis" - for what?

Line 62. Can you please provide some indication as to why a 30 day survival interval was used. Does this interval have some particular significance or was this a question of convenience?

Line 73. How did the population of dogs that met the inclusion criteria compare with the population that did not but that did receive the procedure? By what criteria was it determined that some dogs would and others would not benefit from application of the biomarker assays under examination? To what extent might the selection criteria have biased results?

Line 102. Was cardiac failure the only endpoint that caused an animal to be included in the non-survival group? Does "did not survive until discharge" mean did not survive for 30 days? Same question for dogs that were readmitted and subsequently died - within 30 days?

Table 1. There seems to be something wrong with this table - I don't see anything that appears to be a percentage. The latter only appear in the text.

Line 141. It would seem reasonable to conclude that neither biomarker was able to distinguish between survivors and non-survivors, though without more information on the clinical status of the cases the information provided is difficult to interpret. The correlation coefficients stated are negative for the relationship with length of hospitalization. It would be interesting to see the authors' comment on this finding.

Line 147. If 50% of the survivors were only hospitalized for four days how was follow-up conducted and what constituted grounds for the owners or the referring veterinarian to inform the research group of status? What information was gathered on potential progress during the period after discharge? 

Line 166. This study did not evaluate clinical outcomes beyond 30 days - it would be more accurate to state "30 day survival".

Line 171. Please define "procedural success".

Line 175. Use of the terms inclusion and exclusion criteria here is a little confusing since they are usually used in reference to study design. I think you are referring to the use of the tests to determine selection of suitable surgical candidates, are you not?

Line 176. "… observed 30 day survival rate…"

Wine 176. Insufficient detail is provided in the present paper to draw conclusions from comparison between this study and that of Potter et al. 2024. Inclusion of this previous paper is absolutely essential in the description of the present study, but if any comparisons are to be made much more information needs to be presented concerning how the two investigations proceeded. Some attempt also needs to be made to harmonize or explain differences between definitions and experimental criteria. The issues raised at line 179 appear to apply equally to both centres.

Line 194. It is not clear what the discussion in this paragraph is intended to convey. Comorbidities were higher in the survival group. These may undoubtedly have influenced biomarker levels and survival yet one would expect these to have worked in the opposite direction to that observed. The discussion concerning cTnI levels again seems to be of limited relevance in the context of the paper. Perhaps the group sizes were too small here to draw clear conclusions, but adding comments on more generic aspects of disease processes that influence levels does not add to the paper's ability to comment on mitral valve disease and the application of TEER.

Line 208. Again, the authors are discounting the fact that their correlation coefficient was negative as well as weak.

Line 228. I suggest you consider eliminating this first sentence, since this is in fact something that your study did not achieve. The prediction of outcomes is always complex in clinical medicine - perhaps group size and population here were unable to yield useful information. We have insufficient information in the paper to draw conclusions on operator experience or procedural standardization since no such detail is provided. The reference to long-term follow-up is well-considered and might be a useful topic to address, with definitions, in a revised paper.

Author Response

Response to Reviewer’s comments

We sincerely thank the reviewers and editor for their insightful and constructive feedback, which have significantly improved the manuscript. In response to the concerns raised, we have increased the sample size to allow for a more meaningful comparison with previous studies. Additionally, we have refined our focus from 30-day mortality to procedural safety to better reflect clinical relevance and algin with previous study. We have also expanded our discussion to include post-procedural changes in cardiac size, providing further insight into TEER’s impact. We hope these revisions address the reviewer’s concerns and remain open to any further suggestions for improvement.

Reviewer 2 Report

Comments and Suggestions for Authors

Dear Editor and Authors,

Upon reviewing the manuscript titled "Integrated Biological and Chemical Investigation of Indonesian Marine Organisms Targeting Anti-Quorum Sensing, Anti-Biofilm, Anti-Biofouling and Anti-Biocorrosion Activities" I offer the following peer review comments:

1. The study only included 19 dogs, which is a small sample size. This limits the statistical power and generalizability of the results. For some potential effects, due to the small sample size, significant differences may not be detected. It is recommended to clearly point out this limitation in the conclusion section and emphasize the need for future studies with larger sample sizes to further verify the findings.

2. As a single-center study, there may be selection bias, affecting the external validity of the results. Different centers may have differences in TEER operation techniques and postoperative management, and the data from a single center may not fully represent other centers. It is suggested to mention the necessity of multicenter studies in the discussion section to improve the wide applicability of the research results.

3. Although the inclusion criteria for dogs with MMVD stage C or D and with cTnI and NT-proBNP measurements are clearly defined, it is not clear whether other factors that may affect the results (such as severe arrhythmias) have been excluded. It is recommended to list all the inclusion and exclusion criteria in detail to ensure the homogeneity of the research subjects and reduce the interference of confounding factors.

4. The study results show that NT-proBNP and cTnI levels are not significantly associated with 30-day survival, but the possible reasons for this are not further explored. It is suggested to analyze the dynamic changes of biomarker levels at different survival times (such as 1 week, 2 weeks, 4 weeks), as well as their relationship with postoperative complications. This may help to better understand the potential role of biomarkers in prognosis.

5. It was found that NT-proBNP is weakly correlated with the length of hospital stay, but this phenomenon is not deeply discussed. It is suggested to combine clinical conditions (such as postoperative recovery, complication management, etc.) to analyze the clinical significance of this correlation, and explore whether it is possible to shorten the length of hospital stay by reducing the level of NT-proBNP, or whether this correlation indicates some potential pathological processes.

6. In the discussion section, although comparisons with human study results are mentioned, the comparison with other similar canine TEER studies is not sufficient. It is suggested to compare the results of this study with other published canine TEER studies in detail in terms of biomarker levels, survival rates, complications, etc., and analyze the possible reasons, such as study design, sample size, operation technology, etc.

7. The box plots in Figures 1 and 2 are relatively simple, but detailed legends are not provided, such as the meanings of symbols. It is suggested to improve the legends and annotations of the charts so that readers can more clearly understand the information presented in the charts.

8. The guiding significance of the research results for clinical practice is not fully explained. It is suggested to clearly point out in the discussion and conclusion sections that, although the predictive value of biomarkers is currently limited, clinical decisions still need to comprehensively consider multiple factors such as biomarker levels, clinical symptoms, and imaging examinations to develop individualized treatment plans.

Author Response

We sincerely thank the reviewers and editor for their insightful and constructive feedback, which have significantly improved the manuscript. In response to the concerns raised, we have increased the sample size to allow for a more meaningful comparison with previous studies. Additionally, we have refined our focus from 30-day mortality to procedural safety to better reflect clinical relevance and algin with previous study. We have also expanded our discussion to include post-procedural changes in cardiac size, providing further insight into TEER’s impact. We hope these revisions address the reviewer’s concerns and remain open to any further suggestions for improvement.

Round 2

Reviewer 1 Report

Comments and Suggestions for Authors

vetsci 3421991

I find the paper greatly improved, easier to read and follow and much more succinct. Ambiguity concerning endpoints remains. I make suggestions in relation to this problem in discussion of line 63 of the manuscript, below. I encourage the authors to make these changes and to resubmit - I sincerely believe the changes would greatly enhance the value of the submission.

Line 23 and elsewhere. Please be consistent in terminology when referring to the procedure. You use both "procedure safety" and "procedural safety". The former might be the best choice.

Line 33. “…predictors of procedure…”

Line 63. This may be something left over from the first version of the paper, but you are referring to hospital discharge again as an endpoint yet your notes reference hospitalization duration, which is not in fact stated. This yet again introduces ambiguity. Hospital discharge is also a somewhat ambiguous endpoint without more information. A dog may survive surgery, look good and then rapidly deteriorate, it may improve progressively after surgery, it may show no improvement or it may deteriorate directly after surgery. If hospital discharge is to be used then you have to state criteria by which the dog was discharged - for example, "progressively improving, bright and alert with normal appetite and vital signs and no evidence of deterioration in cardiovascular function". 

Can I recommend that you look closely at your data and identify by plotting variables, an endpoint that is available for the majority of cases and is clinically meaningful. This might be as simple as stable or improving condition over the first week postoperatively, it might be the actual clinical criteria by which you decide that an animal can be discharged. If you do this, be careful to include general information on what might have caused some dogs to have a longer hospitalization than others. This information for the animals that survived to discharge as well as more detail on those that did not survive would provide very useful information for others considering this technique and would enhance the contribution the paper makes to the veterinary medical community. It would eliminate the ambiguity and make the very idea of survival to discharge so much more meaningful.

Line 107. "… included dogs…" - Which begs the obvious question of what else it included, or didn't. Perhaps you mean that the nonsurvival group were dogs that died or were euthanized, i.e., they were never discharged alive.

Line 189. Suggest you delete the words "the biomarkers in dogs such as".

Line 190. "… association with procedure safety, suggesting that these biomarkers may not be of value in the selection of suitable surgical candidates…"

Line 193. Your endpoint must be stated when referring to survival rate. Here you might use survival rate to hospital discharge of 76%.

Author Response

Thank you for the reviewers' thoughtful and constructive feedback. The suggestions have greatly improved the clarity and quality of our manuscript, and we truly appreciate the time and effort put into reviewing our work. We hope that the revisions meet the reviewers' expectations, but we are happy to make any further modifications if needed. Once again, we sincerely appreciate the valuable insights and guidance.

Reviewer 2 Report

Comments and Suggestions for Authors

Authors have revised the manuscript and it could be accepted.

Author Response

(The authors gave the same response as above.)

Round 3

Reviewer 1 Report

Comments and Suggestions for Authors

I am very happy with the revised manuscript. It is succinct, easy to follow and informative.